# Intriguing Properties of Learned Representations

## Abstract

A key feature of neural networks, particularly deep convolutional neural networks, is their ability to "learn" useful representations from data. The very last layer of a neural network is then simply a linear model trained on these "learned" representations. Despite their numerous applications in other tasks such as classification, retrieval, clustering etc., a.k.a. transfer learning, not much work has been published that investigates the structure of these representations or indeed whether structure can be imposed on them during the training process.

In this paper, we study the effective dimensionality of the learned representations by models that have proved highly successful for image classification. We focus on ResNet-18, ResNet-50 and VGG-19 and observe that when trained on CIFAR10 or CIFAR100, the learned representations exhibit a fairly low rank structure. We propose a modification to the training procedure, which further encourages low rank structure on learned activations. Empirically, we show that this has implications for robustness to *adversarial examples* and *compression*.

## 1 Introduction

Among the many successes of deep (convolutional) neural networks, an intriguing aspect has been their ability to generate representations of raw data that are useful in several tasks, usually known as *representation learning*. In the early days of deep learning, it was common to use unsupervised learning models, e.g. auto-encoders or Restricted Boltzmann machines, to *learn* useful representations of complex data (Hinton & Salakhutdinov, 2006; Vincent et al., 2010); more recently, it has been observed that hidden layers of neural networks trained in a completely supervised fashion can also be used as learned representations Zeiler & Fergus (2014); Sermanet et al. (2014); Donahue et al. (2014).

Essentially, for most models trained in a supervised fashion, the vector of activations in the penultimate layer (or at least in a layer close to the output) is a *learned* representation of the raw (for the purposes of this paper *image*) data. The final layer of a neural network is typically simply a multi-class logistic regression model. In this work, we mostly focus on the ResNet-18 and ResNet-50 (He et al., 2016), though we also report some results on VGG-networks (Simonyan & Zisserman, 2014). Although, several aspects of architectures of neural networks have been widely studied in the recent years, there has been little work on understanding the nature of these learned representations.

The learned representations have often been used in other tasks related to classification, retrieval, clustering etc. (often unrelated to the original classification problem) with a good degree of success (a.k.a. transfer learning) (Kiros et al., 2014; Lin & Parikh, 2015). Classically, representation learning has focused on finding low-dimensional structures or independent components through various models, from component analysis to auto-encoders. However, while hidden layers of neural networks are widely used as representations, as far as we are aware, not much work has been published on studying their structure. The current work studies these learned representations; our focus has been on the effective dimensionality, (as opposed to the actual dimension) of these learned representations. A ResNet-18/50 network consists of four ResNet blocks (each of which includes several convolutional layers and skip connections). We consider the dimensionality of the activations obtained at the end of the third and fourth ResNet block. Each data point $\mathbf{x}$ maps to a vector $\mathbf{a} \in \mathbb{R}^m$, where $m$ is the number of units in one of the aforementioned layers; this vector is a *learned* representation of $\mathbf{x}$.

We propose a modification to the training procedure to specifically make the activations (approximately) lie in a low rank space; more precisely, we add a term to the loss that encourages the activations in certain layers to lie in a low-rank affine subspace. The modified training process results in essentially no loss in accuracy (in some cases even shows modest gains) and further enhances the low-rank nature of the learned representations. The modification "adds" *virtual low-rank layers* to the model that ensure that the learned representations roughly lie in a low-rank space. The modified objective function is optimized using an alternate minimization approach, reminiscent of that used in iterative hard thresholding (Blumensath & Davies, 2009) or singular value projection (Jain et al., 2010). Using a naïve singular value thresholding approach would render the training intractable for all practical purposes; we use a column sampling based Nyström method (Williams & Seeger, 2001; Halko et al., 2011) to achieve significant speed-up, though at the cost of not getting the optimal low rank projections. One can view this modified training process as a way to constrain the neural network, though in a way that is very different to the widely used sparsity inducing methods (eg. (Anwar et al., 2017; Wen et al., 2016)) or structurally constrained methods (eg. (Moczulski et al., 2015; Liu et al., 2015)) that seek to tackle the problem of over-parameterization.

Finally, we also investigate the benefits of learning low-rank representations. One obvious benefit is the ability to compress the embeddings when they are used in other applications. The fact that these learned representations (approximately) lie in a low-dimensional (affine) space yields a natural (lossy) compression scheme. Furthermore, we investigate the robustness of networks trained in this fashion to adversarial attacks (Szegedy et al., 2013). Our experimental evaluation shows that networks trained in this fashion are substantially more robust to adversarial attacks than the standard architectures. In our experiments we look at gradient sign methods, its variants (Kurakin et al., 2016; 2017) and DeepFool (Moosavi-Dezfooli et al., 2016). We further perform empirical evaluation showing that when we train SVM classifiers on learned representations (or their low rank projections), the networks trained according to our modified procedure yield significantly more accurate predictions, especially when using very low rank projections of the learned representations, and are in general more robust to adversarial attacks.

**Related Work:** Empirical work by Oyallon (2017) suggests that conditioned on the class, these representations seem to (approximately) lie in a low-rank (affine) space; his work focuses on a different network architecture. Recent (and parallel) work by Lezama et al. (2017) promotes class-conditional low rank embeddings and orthogonality between classes within a single mini-batch to create better discriminative models. They use nuclear norm as a relaxation of rank of a matrix and a loss reminiscent of Qiu & Sapiro (2015) to achieve this and due to their focus on mini-batches, it is unclear if the data as a whole exhibits low-rank structure. Their experiments do not consider adversarial robustness or compression.

From the point of view compression, it is natural to look at methods considering low rank approximations of model parameters (Jaderberg et al., 2014; Denton et al., 2014) as possible alternatives to our algorithm. However, low rank weights do not necessarily create low rank activations due to the non-linearities in neural networks. For similar reasons, reconstructions using auto-encoders also does not exhibit a low-rank structure (See Appendix A).

**Organisation**. The rest of the paper is organized as follows. Section 2 describes the relevant notation, the basic optimization problem and our training procedure. In Section 3, we list our results demonstrating the benefits of the modified training procedure to obtain better low-rank representations and in Section 4, we present results that demonstrate our network's robustness to adversarial attacks.

## 2   LOW RANK PRIOR ON ACTIVATIONS

We define the activation matrix, $A_\ell$, of our network after a certain layer $\ell$; $A_\ell \in \mathbb{R}^{n \times m}$, where $n$ is the number of examples in our training set and $m$ is the dimension of the activation vector after the $\ell^{th}$ layer. By imposing a low rank prior on this matrix, we enforce that the matrix $A_\ell$ is approximately low rank.

We observe that even though enforcing a rank constraint on $A_\ell$ does not obviously increase the sparsity of the model, if we consider a rank $r$ approximation, where $r \ll m$, then there is no need to have more than $r$ units in the subsequent layer. Thus, having an (approximately) low rank representation allows us to compress both the model and the representations with little to no loss in accuracy (as our experimental results demonstrate).

**Optimization problem:** We impose a low rank prior by adding an extra rank regularizor as part of the training objective. We do not regularize the model parameters directly. Instead, we impose constraints on intermediate model outputs as a part of the optimization problem.

Let the neural network be represented by the function $f(\cdot)$, let $X = [\mathbf{x}_1, \cdots, \mathbf{x}_n]^\top$ be the set of training inputs to the model and $Y = [y_1 \cdots y_n]^\top$ be the desired outputs. Consider the network to be composed of two networks, one consisting of layers before the $\ell^{th}$ layer and the other after, i.e. $f(\mathbf{x}_i) = f_\ell^+ \left( f_\ell^- (\mathbf{x}_i) \right)$. Here $f_\ell^+(\cdot)$ represents the network after the $\ell^{th}$ activation layer and $f_\ell^-(\cdot)$ represents the network that produces the $\ell^{th}$ activation layer. Let $f_\ell^-(X) = A_\ell = [\mathbf{a}_1 \cdots \mathbf{a}_n]^\top \in \mathbb{R}^{n \times m}$ represent the *activation matrix* of $n$ data points where $m$ is the dimension of the hidden layer. Our additional low rank constraint requires that $\mathrm{rank}(A_\ell) = r \ll m$.

Let $\mathcal{L}(\theta, \phi; X, Y)$ be the loss function where $f, X, Y$ are as defined above and $f_\ell^+(\cdot)$ and $f_\ell^-(\cdot)$ are parameterized by $\theta$ and $\phi$ respectively. The optimization problem can be written as follows:

$$\theta, \phi = \underset{\theta, \phi}{\mathrm{argmin}} \, \mathcal{L}(\theta, \phi; X, Y) \qquad \text{s.t.} \quad \mathrm{rank}\left(f_\ell^-(X; \phi)\right) = r \qquad \text{(OPT)}$$

A natural approach to this (non-convex) problem would be an alternate minimization algorithm, where the two parts $\mathcal{L}(\theta, \phi; X, Y)$ and $\mathrm{rank}\left(f_\ell^-(X; \phi)\right)$ are optimized alternately. The first of these steps is typically a non-convex optimization problem while the second is a projection step onto a non-convex set which can be solved by performing a singular value decomposition. However, in practice, this approach is infeasible because:

$(i)$ $f_\ell^-(X; \phi)$ is a large matrix and minimizing its rank at every iteration using a singular value decomposition is extremely costly, if not infeasible. (The dimension of this matrix can be as large as $50,000 \times 16,384$ in one of our experimental settings, but in each case is at least $50,000 \times 512$.)

$(ii)$ Minimizing the rank of the corresponding matrix for mini-batches i.e. $f_\ell^-(X_{1:b}; \phi)$ does not guarantee that the rank of $f_\ell^-(X_{1:n})$ will also be small.

Below, we discuss an efficient approach to solve the optimization problem (OPT) by considering some approximations, augmentations and sampling.

**Augmented optimization problem:** To avoid the aforementioned problems, we augment the optimization problem (OPT) by introducing an auxiliary low rank parameter $W \in \mathbb{R}^{m \times m}$ and a bias term $\mathbf{b} \in \mathbb{R}^{m \times 1}$. A rank regularizor term is added to the objective to shift the rank constraint from the activation matrix $A_\ell$ to these auxiliary parameters as explained below. The new augmented optimization problem is as follows:

$$\theta, \phi, W, \mathbf{b} = \underset{\theta, \phi, W, \mathbf{b}}{\mathrm{argmin}} \, \mathcal{L}(\theta, \phi; X, Y) + \lambda_1 \mathcal{L}_c(W, \mathbf{b}; A) + \lambda_2 \mathcal{L}_N(A) \qquad \text{(A-OPT)}$$

$$\text{where,} \quad W \in \mathbb{R}^{m \times m}, \, \mathrm{rank}(W) = r, \, \mathbf{b} \in \mathbb{R}^m, \, A = f_\ell^-(X; \phi)$$

$$\mathcal{L}_c(W, \mathbf{b}; A) = \frac{1}{n} \sum_{i=1}^n \left\| W^\top (\mathbf{a}_i + \mathbf{b}) - (\mathbf{a}_i + \mathbf{b}) \right\|_2^2, \quad \mathcal{L}_N(A) = \frac{1}{n} \sum_{i=1}^n \left| 1 - \|\mathbf{a}_i\| \right|$$

The projection loss $\mathcal{L}_c$ is used to minimize the distance between the activation matrix $A_\ell$ and its affine low rank projection. The bias $\mathbf{b}$ allows for the activation matrix to be translated before projection. The intuition behind $\mathcal{L}_c$ is that if $\mathbf{a}_i \in \mathbb{R}^n$ lies in a (low) $r$-dimensional subspace of $\mathbb{R}^n$, then there is a rank-$r$ matrix $W$ such that $W$ is the identity map on the subspace and maps the rest of $\mathbb{R}^n$ to $\mathbf{0}$. $\mathcal{L}_c$ is a soft version of this and also allows translation by $\mathbf{b}$, i.e. if $A$ is the matrix of activation vectors and $W$ is low-rank, then $WA$ is low-rank; imposing the constraint that $A \approx WA$ forces $A$ itself to be close to low-rank. This is closely related to online PCA. However, setting $A + \mathbf{b}$ close to zero trivially minimizes $\mathcal{L}_c(W, \mathbf{b}; A)$, especially when the activation dimension is large. To prevent this, $\mathcal{L}_N(\cdot)$, which acts as a norm constraint on the activation vector, is introduced to keep the activations sufficiently large.

**Implementing the low rank prior:** Algorithm 1 in Appendix B.1 describes how to apply this low rank regularizor by adding a *virtual* layer called LR-layer after the $\ell^{th}$ layer. This layer is virtual in the sense that it only affects the parameter $W$ and $\mathbf{b}$ which are not directly used to make predictions, but nonetheless the corresponding loss term $\mathcal{L}_c$ does affect the network model parameters through gradient updates. This layer finds the closest affine low rank approximation of $A$ online by estimating a matrix $W$ that minimizes $\mathcal{L}_c$ subject to $\mathrm{rank}(W) = r$.

The rank projection step in Line 16 in Algorithm 1 is executed by a hard thresholding operator $\Pi_r^{\text{rank}}(W)$, which finds the best $r$-rank approximation of $W$. Essentially, $\Pi_r^{\text{rank}}(W)$ solves the following optimization problem, which can be solved using a singular value decomposition (SVD).

$$\Pi_r^{\text{rank}}(W) = \underset{\text{rank}(Z)=r}{\text{argmin}} \ \|W - Z\|_F^2 \tag{1}$$

**Handling large activation matrices:** Singular Value Projection (SVP) introduced in Jain et al. (2010) is an algorithm for rank minimization under affine constraints. In each iteration, the algorithm performs gradient descent on the affine constraints alternated with a rank-k projection of the parameters and it provides recovery guarantees under weak isometry conditions. However, the algorithm has a complexity of $O(mnr)$ where $m, n$ are the dimensions of the matrix and $r$ is the desired low rank. Faster methods for SVD for sparse matrices are not applicable as the matrices in our case are not necessarily sparse. We use the ensembled Nyström method (Williams & Seeger, 2001; Halko et al., 2011; Kumar et al., 2009a) to boost our computational speed at the cost of accuracy of the low rank approximation. It is essentially a sampling based low rank approximation to a matrix. The algorithm is described in detail in the Appendix B.2. Though the overall complexity for projecting $W$ still remains $O(m^2 r)$, the complexity of the hard-to-parallelize SVD step is now $O(r^3)$, while the rest is due to matrix multiplication.

The theoretical guarantees of Nyström hold only when the weight matrix of the LR-layer is symmetric and positive semi-definite (PSD) before each $\Pi_r^{\text{rank}}(\cdot)$ operation; this restricts the projections allowed in our optimization, but empirically this does not seem to matter. We know that a symmetric diagonally dominant real matrix with non-negative diagonal entries is PSD. With this motivation, the matrix $W$ is smoothened by repeatedly adding $0.01\mathcal{I}$ until the SVD algorithm converges[1]. This is a heuristic to make the matrix well conditioned (as well as diagonally dominant) and it helps in the convergence of the algorithm empirically.

**Symmetric Low Rank Layer:** The Nyström method requires the matrix $W$ of the LR-layer to be symmetric and PSD (SPSD), however, gradient updates may make the matrix parameter non-SPSD, even if we start with an SPSD matrix. Reparametrizing the LR-layer fixes this issue; the layer is parameterized using $W_s$ (to which gradient updates are applied), but the layer projects using $W = (W_s + W_s^\top)/2$. After the rank projection is applied to the (smoothed version of) $W$, $W_s := \Pi_r^{\text{rank}}(W)$ is an SPSD matrix (using Lemma 1 in Appendix B.3). As a result the updated $W$ is also SPSD. This layer also has a bias vector $\mathbf{b}$ to be able to translate the activation matrix before performing the low rank projection.

**Model Compression:** As a consequence of forcing the activations of the $\ell^{th}$ layer of the model to lie in a very low dimensional subspace with minimal reconstruction error and loss in accuracy, our experiments suggest that a simpler model can replace the latter parts of the original model without significant reduction in accuracy. Essentially, we can replace $f_\ell^+ : \mathbb{R}^m \to \mathbb{R}^c$ with $g : \mathbb{R}^{m'} \to \mathbb{R}^c$ where $m' \ll m$ and $g$ is a much smaller model than $f_\ell^+$. Our experiments in Table 2 show empirical evidence of this.

## 3 EXPERIMENTS

In this section we look at experimental validation of the various properties of our networks. As our primary training objective is to reduce the rank of the activations, we use the *variance ratio*, defined as $\text{VR}_r(A) = \frac{\sum_{i=1}^r \sigma_i^2}{\sum_{i=1}^p \sigma_i^2}$, where $\sigma_1, \ldots, \sigma_p$ are the singular values of $A$ in non-increasing order, to estimate the variance captured by the first $r$ out of $p$ singular values.

We consider the following models in our experiments:

(a) ResNet 1-LR - This model contains one LR-layer, located immediately before the last fully connected (FC) layer. For a ResNet-18, the incoming activations have 512 units whereas for a ResNet-50 the incoming activations has a dimension of 2048.

(b) ResNet 2-LR - This model contains two LR-layers. The first LR-layer is positioned before the fourth ResNet block, where the incoming activations have a dimension of $16,384$ and the second LR-layer is before the FC layer as in ResNet 1-LR.

---

[1]The computation of the singular value decomposition sometimes fail to converge if the matrix is ill-conditioned

(c) VGG19 2-LR - This includes two LR-layers in the VGG-19 model. The VGG model has three FCs after 16 convolution layers with the LR-layers present before the first and the third FC layers in the model.

(d) X-MAXG - This is a hybrid model where learned representations are generated for the input using the model "X" and then a maximum margin classifier is learned on these representations. The particular layer from which the representation is extracted will be clear from the context.

(e) Bottle-LR - This model has a bottle neck layer which is essentially a fully connected layer without any non-linear activation. Moreover, the weight matrix of the $\mathbf{W} \in \mathbb{R}^{m \times m}$ is parametrized by $\mathbf{W}_l \in \mathbb{R}^{m \times r}$ where $\mathbf{W} = \mathbf{W}_l \mathbf{W}_l^\top$ such that it is always low rank by definition.

Some other models that are comparable to this work are listed in Appendix A. While one of the main objectives has been to design an augmentation to the training algorithm instead of changing the model itself (note that the extra weight matrices can be removed from the model after training is complete), we also consider Bottle-LR which includes an extra layer. The Bottle-LR layers have the same dimensions and are placed at the same position as the LR layers. As the experiments below suggest, the Bottle-LR model does not show the same benefits our LR model shows and performs no better than the N-LR model (indeed worse sometimes). One of the main difference in the training dynamics between this model and our model is that an explicit bottleneck layer has a *multiplicative* effect on the gradients propagated backward in the network, whereas, in our approach they only have an *additive* effect. Although this is somewhat speculative at this stage, we believe the relative "smoothness" of our approach is the reason why it seems to have better performance on these other tasks.

In this paper, we used the datasets CIFAR10 and CIFAR100. As a side note, we observed that the target rank is not an essential hyper-parameter as the training enforces a much lower rank than what is set. Further details on the training procedure and the X-MAXG model is provided in Appendix C. Apart from these, we use the term N-LR to represent models without LR-layers.

We perform three kinds of experiments in this section, with additional experiments in Appendix C.3 and C.4. We first show that our imposed constraints do not have any significant impact on the test accuracy of the model. Then we go on to verify that our training indeed achieves its objective of developing a stronger low rank structure. Finally we demonstrate that our training algorithm provides a natural scheme for compressing our models and that low dimensional projections of our embeddings, with a size of less than 2% of the original embeddings, can be used for classification with a significantly higher accuracy than similar sized projections of embeddings from an N-LR model. In Section 4, we provide empirical evidence in support of adversarial robustness of our models.

**Impact on test accuracy:** In this experiment, we check if the additional constraints on the training method have any significant effect on the performance of the model. We observe, as listed in Table 1, that the additional penalties impose no significant loss in accuracy. In some cases, we even observe small gains in performance.

| Models | Test Accuracy |
|---|---|
| ResNet 1-LR | 92.8 |
| ResNet 2-LR | 92 |
| VGG19 2-LR | 89.8 |
| ResNet N-LR | 92.5 |
| VGG19 N-LR | 89.1 |
| Bottle-2LR | 92 |

(a) ResNet18 on CIFAR-10

| Models | Coarse Label | Fine Label | $\text{VR}_{20}\,(\cdot)$ |
|---|---|---|---|
| ResNet 1-LR | 78.1% | 48% | 0.97 |
| ResNet N-LR | 75.6% | 52% | 0.90 |
| Bottle-1LR | 76% | 38% | NA |

(b) ResNet50 on CIFAR-100

Table 1: Test Accuracy

To see whether the learned representations can be used in a different task, we conduct a transfer learning exercise where embeddings generated from a ResNet-50 model, trained on the coarse labels of CIFAR-100, are used to predict the fine labels of CIFAR-100. A set of ResNet-50-MAXG classifiers are trained for this purpose on these embeddings. (see Appendix C.1 for further details). We can see in Table 1(b) that the LR model suffers a small loss of 4% in accuracy as compared to the N-LR model when it's embeddings are used to train a max margin classifier for the task of predicting the fine labels. However, it is to be noted that the accuracy of the LR model actually increases when the max margin classifier is trained on it to do the original task i.e. classifying the coarse labels. On

the other hand, the Bottle-LR model suffers a loss of $14\%$ in accuracy compared to N-LR model and does not show any advantage in the original task either.

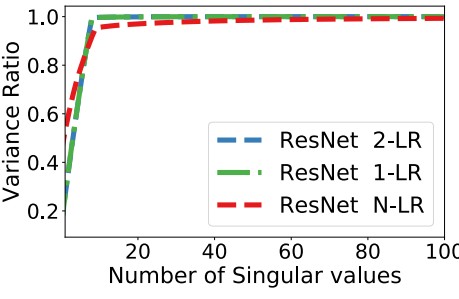

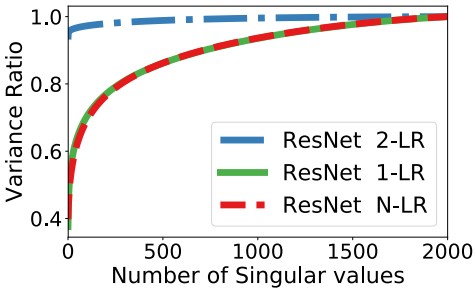

(a) Activations after last ResNet block.

(b) Activations before last ResNet block

Figure 1: Variance Ratio captured by varying number of Singular Values

**Effective rank of activations:** Figure 1(a) shows the variance ratio captured by varying numbers of singular values in the activations before the last FC layer. In this case, every model shows that the *effective* rank of the activations is 10 as there is a sharp elbow in the plot after 10 singular values. However, the LR-model has almost negligible leakage of variance compared to the N-LR model. Similar experiments for VGG is found in the appendix (Figure 5). .

Figure 1(b) shows the variance ratio captured by varying numbers of singular values for the activations before the fourth ResNet block. In this case the activation vector has a dimension of $16,384$ and the use of Nyström method is necessary for computational feasibility. ResNet 2-LR is the only model that has a LR-layer in that position and Figure 1(b) shows that it is the only model that shows a low rank structure on that layer.

While there are various techniques developed that induce sparsity on weights or activations in neural networks, it is important to point out that sparsity doesn't necessarily lead to low rank eg. an identity matrix. Empirically, in ResNet 1-LR, the activations before the $4^{th}$ ResNet block are much less explained by a small number of singular values despite the high level of sparsity ($39\%$) whereas the activations after the 4th ResNet block are explained by about 10 singular values though it is $95\%$ dense.

**Validity of low dimensional embeddings:** The previous experiment shows that our training algorithm introduces a low rank structure in the activations. However, it does not show any evidence of the discriminative power of these low rank embeddings. In these experiments, we use both CIFAR-10 and CIFAR-100 with ResNet-18 and ResNet-50 respectively to show that these embeddings and their projections onto low dimensional spaces are effective at discriminative tasks and are better at it than models without the LR-layer.

| Model | Emb-dim | Acc |
|---|---|---|
| ResNet-50-LR | 2048 | 78.1 |
| ResNet-50 | 2048 | 75.6 |
| ResNet-50-LR | 10 | 76.5 |
| ResNet-50 | 10 | 68.4 |
| ResNet-50-LR | 5 | 72 |
| ResNet-50 | 5 | 48 |

(a) Representation from before the FC layer of a ResNet-50 trained on CIFAR-100.

| Model | Emb-dim | Acc |
|---|---|---|
| ResNet18-2-LR | $16,384$ | $91.14\%$ |
| ResNet18 | $16,384$ | $90.7\%$ |
| ResNet18-2-LR | 20 | $88.5\%$ |
| ResNet18 | 20 | $76.9\%$ |
| ResNet18-2-LR | 10 | $75\%$ |
| ResNet18 | 10 | $61.7\,\%$ |

(b) Representation from before the last ResNet block of ResNet-18 trained on CIFAR-10.

Table 2: Accuracy of low dimensional projections of learned representations at discriminative tasks.

Table 2(a) shows that even with decreasing embedding dimension, the LR model is able to preserve its accuracy better than the N-LR model. Even with a $5$-dimensional embedding, the LR model looses only $6\%$ in accuracy, but the N-LR model looses $27\%$.

---

[1]Leakage of variance for $k$ singular values is mathematically defined as $1 - \mathsf{VR}_k\left(\cdot\right)$

The results of the next experiment, listed in Table 2(b), show the capability of our algorithm to compress the *model* itself. The entire fourth ResNet block along with the last FC layer (referred to as $f_\ell^+$ and containing 8.4M parameters) can be replaced by a smaller linear model which has only 0.02 times the number of parameters as $f_\ell^+$. This yields a significant reduction in model size in exchange for a slight drop in accuracy ($< 1\%$). The second benefit is that as the low dimensional embeddings still retain most of the *discriminative* information, the inputs fed to the linear model have a small number of features.

## 4 ADVERSARIAL ATTACKS

(Szegedy et al., 2013) showed that adding adversarial perturbations to inputs to machine learning models, that otherwise perform well on a test set, can often make them suffer a high misclassification rate. To express this formally, consider a machine learning model $M : X \to Y$ where $X$ is the input space and $Y$ is the target label space. It is possible to add an adversarial noise $\delta$ to $\mathbf{x}_i \in X$ such that though $M(\mathbf{x}_i)$ predict $\mathbf{y}_i$, but $M(\mathbf{x}_i + \delta)$ predicts a label other than $\mathbf{y}_i$ with high confidence, despite $\mathbf{x}_i$ and $\mathbf{x}_i + \delta$ being perceptually indistinguishable to a human. Various methods (Szegedy et al., 2013; Goodfellow et al., 2014; Kurakin et al., 2017; Moosavi-Dezfooli et al., 2016) have been proposed in recent years for constructing adversarial perturbations. In this section, our experiments show that LR models are more robust than N-LR models against these adversarial perturbations.

In this section, $\mathbf{x}_d$ refers to an example drawn from the data distribution and $\mathbf{x}_a$ denotes the adversarially perturbed version of $\mathbf{x}_d$. For vectors $\mathbf{z}$ and $\mathbf{x}$, let $\mathrm{clip}_{\mathbf{x},\epsilon}(\mathbf{z})$ denote the element-wise clipping of $\mathbf{z}$, with $z_i$ clipped to the range $[x_i - \epsilon, x_i + \epsilon]$.

**Adversarial attacks:** We look at three different attacks in our experiments: (i) Iterative Fast Sign Gradient Method (Iter-FSGM) (Kurakin et al., 2016; Madry et al., 2018) (ii) The Iterative Least Likely Class Method (Iter-LL-FSGM) (Kurakin et al., 2017) (iii) DeepFool (Moosavi-Dezfooli et al., 2016). The reader may refer to Appendix D.1 for further details on the attacks. The iterative fast sign gradient method (Iter-FSGM) is essentially equivalent to the Projected Gradient Descent Method on the negative loss function (PGD, Madry et al. (2018)).

Further, due to the low rank constraints, the model could implicitly be enforcing gradient masking, which might result in the construction of weaker attacks. Hence, to be fair in our comparison, we also consider *black box* versions of each of the aforementioned attacks. While the adversarial noise in a white box attack is constructed using the model being attacked, the noise in a black box attack is constructed using a different model (in our case the N-LR model). We use the standard normalized $L_2$ dissimilarity measure $\rho = \sum_i \frac{\|\mathbf{x}_a - \mathbf{x}_d\|_2}{\|\mathbf{x}_d\|_2}$ to measure the magnitude of this noise.

**Robustness to Adversarial Attacks:**

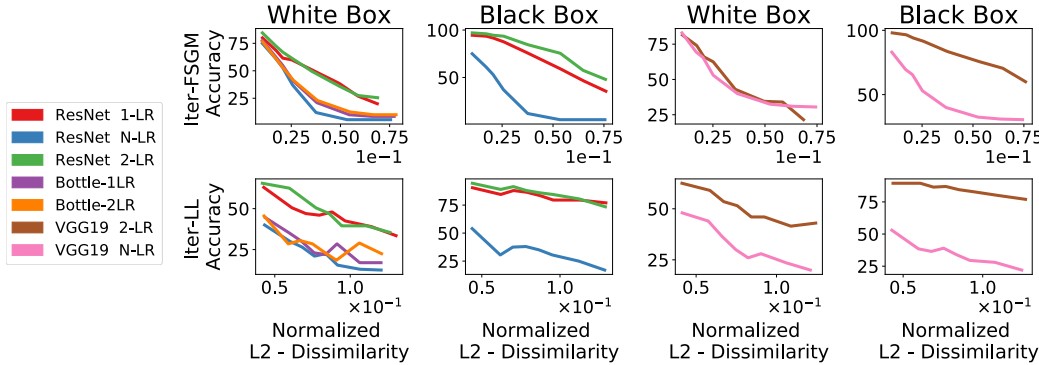

Figure 2: Adversarial accuracy plotted against magnitude of perturbation (measured with $\rho$).

Here we compare across models, the change in accuracy of classifying adversarial examples with respect to the amount of noise added. In line with the experiments in Kurakin et al. (2017), the

noise is added for a pre-determined number of steps. Figure 2 shows that as the noise increases, the accuracy of N-LR models decrease much faster than the LR models.

Specifically, to reach an adversarial mis-classification rate of $50\%$, our models require about twice the noise as a normal model or Bottle-LR model. We also observe that black box attacks are not very successful when multi-step (i.e. iterative) attacks are performed [2]. In short, for *all* kinds of attacks we tried, LR models consistently perform better than N-LR models.

Our next experiment is in line with the experiments conducted in Moosavi-Dezfooli et al. (2016). Table 3 lists the average minimum perturbation (measured with $\rho$) required to make the classifier mis-classify more than $99\%$ of the adversarial examples, constructed from a uniformly sampled subset of the test set. Appendix D.2 describes the setup in further detail.

| | Model | $\rho$ [DeepFool ] | $\rho$ [Iter-LL-FSGM ] | $\rho$ [Iter-FSGM ] |
|---|---|---|---|---|
| | ResNet N-LR | $1.6 \times 10^{-2}$ | $2.4 \times 10^{-2}$ | $2.1 \times 10^{-2}$ |
| White Box | ResNet 1-LR | $1.7 \times 10^{-1}$ | $1.1 \times 10^{-1}$ | $6.0 \times 10^{-2}$ |
| | ResNet 2-LR | $\mathbf{1.8 \times 10^{-1}}$ | $\mathbf{9.8 \times 10^{-2}}$ | $\mathbf{7.6 \times 10^{-2}}$ |
| Black Box | ResNet 1-LR | $4.7 \times 10^{-2}$ | $1.8 \times 10^{-1}$ | $5.6 \times 10^{-2}$ |
| | ResNet 2-LR | $\mathbf{5.5 \times 10^{-2}}$ | $\mathbf{2.0 \times 10^{-1}}$ | $\mathbf{7.5 \times 10^{-2}}$ |

Table 3: Perturbation required for Adversarial Misclassification

An interesting observation is that the values of $\rho$, here in Table 3, are lower than in Figure 2 though the attacks have a higher rate of success. To explain this behaviour, we show empirical evidence to indicate that an attack that adds noise for a fixed number of steps (Kurakin et al., 2017; 2016) to the input is significantly weaker than one that stops on successful misclassification (See Appendix D.3). However, even under this scheme of attacks, our models perform better than N-LR models as LR models require 4 to 11 times the amount of noise required by N-LR models to be fooled by adversarial attacks.

We also look at some of the adversarial images generated by DeepFool in Figure 3. We observe that it is immediately clear that the adversarial images are different from the original images in the case of LR models whereas it is not so apparent in the case of N-LR models.

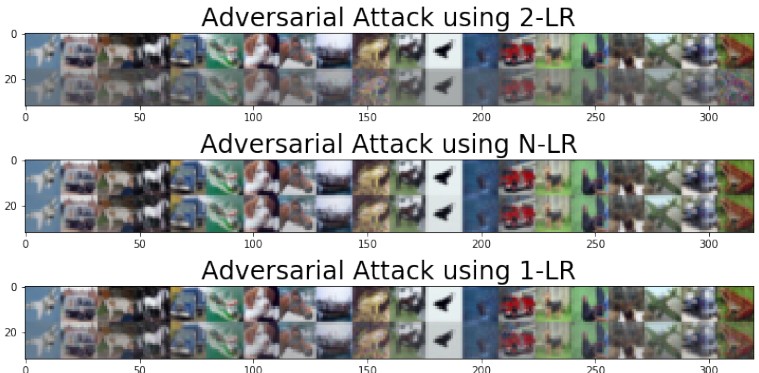

Figure 3: For each model, original images are on the top row and the images generated by DeepFool are below.

In order to gain some intuition into this behaviour of robustness against adversarial attacks, we look into the property of *noise stability* of networks towards adversarial noise. Arora et al. (2018) gives empirical evidence for deep networks being stable towards injected Gaussian noise and uses a variation of this noise stability property to derive more realistic generalization bounds. We believe

---

[2]Conducting single step black-box attacks with large $\epsilon$ were more successful than the results shown in Figure 2. However, even for these, LR models are more robust than N-LR ones, and the images produced are highly perturbed and are not consistent with the rest of the experiments.

that learning a model that zeroes out irrelevant directions in the learned representations reduces the ability to find adversarial perturbations that can affect the output of the model. In Figure 4, we show empirically that LR networks are more noise-stable to adversarial noise than non-LR networks. On the $x$-axis is the normalized $L_2$ dissimilarity score in the input space i.e. $\|\mathbf{x} + \eta\|^2/\|\mathbf{x}\|^2$ and on the $y$-axis the corresponding quantity in the representation space i.e. $\left\|f_\ell^-(\mathbf{x} + \eta)\right\|^2/\left\|f_\ell^-(\mathbf{x})\right\|^2$. The representations here are taken from before the last fully connected and softmax layer. As our experiments suggest, the LR model significantly attenuates the adversarial perturbation in the representation space thus making it harder to fool the softmax classifier.

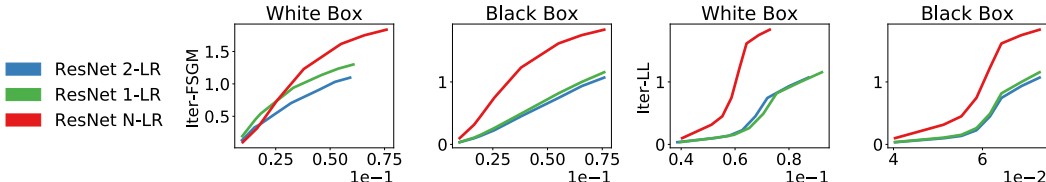

Figure 4: Adversarial Perturbation in Input Space and Perturbation in Representation Space

**Max Margin Classifiers:** Finally, we show that X-MAXG models are significantly more robust to adversarial attacks than the corresponding "X" models. Also, as seen in Table 4, X-MAXG models with LR-layers are more robust than X-MAXG models without LR-layers against adversarial attacks. Our experimental setup is explained in detail in Appendix D.4. We can see that a X-MAXG model with a LR-layer correctly classifies $50\%$ of the examples that had fooled the original classifier while for a similar amount of noise, the X-MAXG model without the LR-layer has negligible accuracy.

|  | Model | DeepFool | Iter-LL-FSGM | Iter-FSGM |
|---|---|---|---|---|
| White Box | ResNet18 N-LR-MAXG | 0.01 | 0.04 | 0.02 |
|  | ResNet 1-LR-MAXG | 0.38 | 0.35 | 0.48 |
|  | ResNet 2-LR-MAXG | 0.43 | 0.55 | 0.55 |
| Black Box | ResNet 2-LR-MAXG | 0.29 | 0.31 | 0.33 |
|  | ResNet 1-LR-MAXG | 0.44 | 0.50 | 0.48 |

Table 4: Accuracy of classification of adversarial examples by Max Margin Classifiers.

## 5 CONCLUSION

In this paper, we designed an algorithm that encourages the learned representations obtained by training deep neural networks for supervised tasks to lie in an approximately low rank (affine) subspace. This is achieved by augmenting the networks with *virtual* LR-layers and modifying the training objective. In order to make our algorithm computationally feasible for large networks, we proposed and implemented certain approximation techniques. Our experiments show that our algorithm successfully enforces an approximate low rank behaviour and that these learned representations have some intriguing properties. We conducted a wide range of experiments to investigate these properties and report, among other things, that (i) max-margin classifiers trained on these representations have more discriminatory power than ones trained on representation from models without LR-layers, (ii) our models are more robust to a variety of adversarial attacks than models without LR-layers, and (iii) LR models significantly attenuate the effects of perturbations introduced in the input space on the representation space (iv) replacing large parts of our models with smaller models results in a negligible drop in accuracy, thereby providing a compression scheme, It is commonly believed that large sparse feature spaces are better for deep models. What we propose here is the idea that while the features themselves need not be sparse, the existence of a basis in which the feature vectors have a sparse representation can provide benefits. To the best of our knowledge, investigating properties of representations learned from deep supervised NNs, and the possibility of modifying training procedures to obtain desirable properties in said representations, has remained relatively unexplored. This work shows that these representations possess some intriguing properties, which may well be worthy of further investigation.

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

## A  ALTERNATIVE ALGORITHMS

### LOW RANK WEIGHTS

With respect to compression, it is natural to look at low rank approximations of network parameters (Denton et al., 2014; Jaderberg et al., 2014). By factorizing the weight matrix/tensor $W$, for input $x$, we can get low rank *pre-activations* $Wx$. This however does not lead to low rank *activations* as demonstrated both mathematically (by the counter-example below) and empirically.

**Mathematical Counter-Example:** Consider a rank 1 *pre-activation* matrix $A$ and its corresponding *post-activation*(ReLU) matrix as below. It is easy to see that the rank of *post-activation* has increased to 2.

$$A = \left[ \begin{array}{ccc} 1 & -1 & 1 \\ -1 & 1 & -1 \end{array} \right] \qquad \text{Relu}(A) = \left[ \begin{array}{ccc} 1 & 0 & 1 \\ 0 & 1 & 0 \end{array} \right]$$

**Empirical Result:** In order to see if techniques for low rank approximation of network parameters like Denton et al. (2014) would have produced low rank activations, we conducted an experiment by explicitly making the *pre-activations* low-rank using SVD. Our experiments showed that inspite of setting a rank of 100 to the *pre-activation* matrix, the *post-activation* matrix had full rank. Though all but the first hundred singular values of the *pre-activation* matrix were set to zero, the *post-activation* matrix's $101^{st}$ and $1000^{th}$ singular values were 49 and 7.9 respectively, and its first 100 singular values explained only $94\%$ of the variance.

We try to explain the above empirical results as follows: Theoretically, a bounded activation function lowers the Frobenius norm of the *pre-activation* matrix i.e. the sum of the squared singular values. However, it also causes a smoothening of the singular values by making certain 0 singular values non-zero to compensate for the significant decrease in the larger singular values. This leads to an increase in rank of the *post-activation* matrix.

### BOTTLENECK LR LAYER

**Bottleneck Layer:** It is easy to see that the effective dimension of the representation of an input, obtained after passing through a bottleneck layer (like an auto-encoder), will not be greater than the dimension of the bottleneck layer itself. However, due to the various non-linearities present in the network, while the representation is guaranteed to lie in a low dimensional manifold it is not guaranteed to lie in a low rank (affine) subspace.

**LR bottleneck:** Another alternative is to include the low-rank projection and reconstruction as part of the network instead of as a regularizor so that the LR-layer is an actual layer and not a *virtual* layer. We have indeed experimented with this setup and observed that this often made the training very unstable. Also, if one were to add this bottleneck as a fine-tuning process, the test accuracy of the network decreases by a much higher extent than it does for our method.

## B ALGORITHMIC DETAILS

### B.1 ALGORITHM FOR LR LAYER

Algorithm 1 below lists the forward and the backpropagation rules of the LR-layer.

---

**Algorithm 1** LR Layer

---

1: **Input:** Activation Matrix $A$, Grad_input $g$
2:
3: **Forward Propagation**
4: $Z \leftarrow W^\top (A + \mathbf{b})^3$      ▷ Compute the affine *Low rank* projection
5: **Output :** $A$      ▷ Output the original activations for the next layer
6:
7: **Backward Propagation**
8: $D_1 \leftarrow \frac{\lambda_1}{n} \|Z - (A + \mathbf{b})\|_2^2$      ▷ Computes the reconstruction loss $\mathcal{L}_c$
9: $D_2 \leftarrow \frac{\lambda_2}{n} \sum_{i=0}^{n-1} |\mathbf{1} - \|\mathbf{a_i}\||$      ▷ Computes the loss for the norm constraint $\mathcal{L}_N$
10: $D \leftarrow D_1 + D_2$
11: $g_W \leftarrow \frac{\partial D}{\partial W}, g_i \leftarrow g + \frac{1}{n} \sum_{i=0}^{n-1} \frac{\partial D}{\partial \mathbf{a_i}}$
12: **Output :** $g_i$      ▷ Outputs the gradient to be passed to the layer before
13:
14: **Update Step**
15: $W \leftarrow W - \lambda g_W$      ▷ Updates the weight with the gradient from $D$.
16: $W \leftarrow \Pi_k^{\mathrm{rank}}(W)$      ▷ Hard thresholds the rank of $W$

---

### B.2 ENSEMBLED NYSTRÖM METHOD

Let $W \in \mathbb{R}^{m \times m}$ be a symmetric positive semidefinite matrix (SPSD). We want to generate a matrix $W_r$ which is a r-rank approximation of $W$ without performing SVD on the full matrix $W$ but only on a principal submatrix[4] $Z \in \mathbb{R}^{l \times l}$ of $W$, where $l \ll m$. We sample $l$ indices from the set $\{1 \cdots m\}$ and select the corresponding columns from $W$ to form a matrix $C \in \mathbb{R}^{m \times l}$. In a similar way, selecting the $l$ rows from $C$ we get $Z \in \mathbb{R}^{l \times l}$. We can rearrange the columns of $W$ so that

$$W = \begin{bmatrix} Z & W_{21}^T \\ W_{21} & W_{22} \end{bmatrix} \qquad C = \begin{bmatrix} Z \\ W_{21} \end{bmatrix}$$

According to the Nyström approximation, the low rank approximation of $W$ can be written as

$$W_r = C Z_r^+ C^T \tag{2}$$

where $Z_r^+$ is the pseudo-inverse of the best $r$ rank approximation of $Z$. Hence, the entire algorithm is as follows.

- Compute $C$ and $Z$ as stated above.

- Compute the top $r$ singular vectors and values of $Z$ : $U_r, \Sigma_r, V_r$.

- Invert each element of $\Sigma_r$ as this is used to get the Moore pseudo-inverse of $Z_r$.

- Compute $Z_r^+ = U_r \Sigma_r^{-1} V_r$ and $W_r = C Z_r^+ C^T$.

Though by trivial computation, the complexity of the algorithm seems to be $O(l^2 r + ml^2 + m^2 l) = O(m^2 r)$ (In our experiments $l = 2r$), it must be noted that the complexity of the SVD step is only $O(k^3)$ which is much lesser than $O(m^2 r)$ and while matrix multiplication is easily parallelizable, parallelization of SVD is highly non-trivial and inefficient.

To improve the accuracy of the approximation, we use the ensembled Nyström sampling based methods (Kumar et al., 2009a) by averaging the outputs of $t$ runs of the Nyström method. The

---

[3]$\mathbf{b} + A$ is computed by adding $\mathbf{b}$ to every row in $A$
[4]A principal submatrix of a matrix $W$ is a square matrix formed by removing some columns and the corresponding rows from $W$ (Meyer, 2000)

$l$ indices for selecting columns and rows are sampled from an uniform distribution and it has been shown (Kumar et al., 2009b) that uniform sampling performs better than most sampling methods. **Theorem 3** in Kumar et al. (2009a) provides a probabilistic bound on the Frobenius norm of the difference between the exact best r-rank approximation and the Nyström sampled r-rank approximation.

### B.3 LEMMA 1

**Lemma 1.** *If $X \in \mathbb{R}^{m \times m}$ is a SPSD matrix and $X_r \in \mathbb{R}^{m \times m}$ is the best Nyström ensembled, column sampled r-rank approximation of $X$, then $X_r$ is SPSD as well. (Proof in Appendix B.3)*

*Proof.* By the Construction of the Nyström SVD algorithm, we know that $X_r = CW_r^+ C^T$. We will first show that $W_r^+$ is a symmetric matrix.

We know that $X$ is SPSD. Let $I$ be a sorted list of distinct indices such that $|I| = l$. Then by construction of $W$,

$$W_{i,j} = X_{I[i],I[j]}$$

Hence, as $X_{I[i],I[j]} = X_{I[j],I[i]}$, $W$ is symmetric.

At this step, our algorithm adds $\delta \cdot I$ to $W$ where $\delta \geq 0$. It is easy to observe that $W + \delta \cdot \mathcal{I}$ is positive semidefinite.

Consider a vector $a \in \mathbb{R}^{|X|}$. Create a vector $\bar{a} \in \mathbb{R}^m$ where

$$\bar{a}_i = \begin{cases} 0 & \text{if } i \notin I \\ a_i & \text{o.w.} \end{cases}$$

$$a^\top (W + \delta \cdot \mathcal{I}) \, a = \bar{a}^\top X \bar{a} + \delta \cdot a^\top \mathcal{I} a \geq 0 + \delta \|a\|^2 \geq 0 \tag{3}$$

Let $W + \delta \mathcal{I}$ be the new $W$ and (3) shows that $W$ is positive semidefinite.

Now we will show that $X_r$ is symmetric as well. As $W$ is symmetric, there exists an orthogonal matrix $Q$ and a non-negative diagonal matrix $\Lambda$ such that

$$W = Q \Lambda Q^T$$

We know that $W_r = Q_{[1:r]} \Lambda_{[1:r]} Q_{[1:r]}^T$ and $W_r^+ = Q_{[1:r]} \Lambda_{[1:r]}^{-1} Q_{[1:r]}^T$.
Hence,

$$
\begin{aligned}
X_r &= CW_r^+ C^T \\
&= CQ_{[1:r]} \Lambda_{[1:r]}^{-1} Q_{[1:r]}^T C^T \\
X_r^T &= (CQ_{[1:r]} \Lambda_{[1:r]}^{-1} Q_{[1:r]}^T C^T)^T \\
&= CQ_{[1:r]} \Lambda_{[1:r]}^{-1} Q_{[1:r]}^T C^T \\
&= X_r
\end{aligned}
$$

$\therefore X_r$ is symmetric. We can also see that the $X_r^T$ is positive semi definite by pre-multiplying and post multiplying it with a non-zero vector and using the fact that $W_r^+$ is positive semi-definite.

$\square$

## C    EXPERIMENTAL DETAILS

We used a single NVIDIA Tesla P40 GPU for training our networks. All our experiments were run with a learning rate of $0.01$ for 500 epochs with a batch size of 512. The hyper-parameter $l$ in the Nyström method was set to double of the target rank. The model was pre-trained with SGD until the training accuracy reached $65\%$ and then Algorithm 1 was applied. The rank cutting operation was performed every 10 iterations.

The target rank for the LR-layers, which were placed before the FC layers was set to $50$ while the target rank for the LR-layer before the last ResNet block was set to 1000. However, experiments suggest that this hyper-parameter is not very crucial to the training process as the training procedure converged the effective rank of the *activation matrix* to a value lesser than the designated target ranks.

The linear classifier in **X-MAXG** is trained using SGD with hinge loss and $L_2$ regularization with a coefficient of $0.01$. The learning rate is decreased per iteration as $\eta_t = \frac{\eta_0}{(1+\alpha t)}$ where $\eta_0$ and $\alpha$ are set by certain heuristics [5].

### C.1    IMPACT ON TEST ACCURACY

Table 1(b) shows the result of a transfer learning exercise where we trained two sets of ResNet-50-MAXG classifiers. First, two ResNet-50 models were trained with and without the LR-layer respectively on the coarse labels of CIFAR-100. Then, 2048 dimensional embeddings were extracted from after the fourth ResNet block using the train set and the test set of CIFAR 100. The embeddings from the train set were used to train two groups (each group having two classifiers- one for each ResNet-50 model) of max-margin linear classifiers with the same hyper-parameters as described above. While the first group was trained on the coarse labels of CIFAR-100, the second set was trained on the fine labels. The test accuracy of these four classifiers are then reported in Table 1(b) on the corresponding labels.

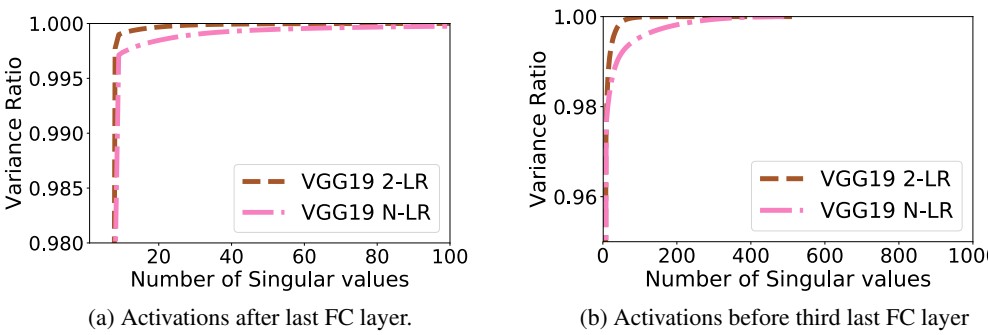

(a) Activations after last FC layer.    (b) Activations before third last FC layer

Figure 5: Variance Ratio captured by varying number of Singular Values

**Effective rank of activations for VGG:**

### C.2    VALIDITY OF LOW DIMENSIONAL EMBEDDINGS

In the first experiment, reported in Table 2(a), we trained two ResNet-50-MAXG models -with and without the LR-layer respectively- on the 20 super-classes of CIFAR-100. As our objective here is to see if the embeddings and their low dimensional projections could be effectively used for discriminative tasks, we used PCA, with standard pre-processing of scaling the input, to project the embeddings onto a low dimensional space before training a linear maximum margin classifier on it.

The experiments in Table 2(b) were run with ResNet-18 on CIFAR-10. Two ResNet-18-MAXG classifiers - with and without the LR-layer respectively- were trained on CIFAR-10. The representations were obtained from before the fourth ResNet block and had a dimension of 16,384. Similar

---

[5]https://goo.gl/V995mD

to the previous experiment, we used PCA, with standard pre-processing, to gather low dimensional projections before training linear max margin classifiers on it.

| Model | Emb-dim | Acc |
|-------|---------|-----|
| VGG19-2LR | 512 | 89.8 |
| VGG19-NLR | 512 | 89.7 |
| VGG19-2LR | 20 | 89.85 |
| VGG19-NLR | 20 | 89.78 |
| VGG19-2LR | 10 | 89.79 |
| VGG19-NLR | 10 | 89.65 |

Table 5: Representation from before the third last FC layer of a VGG19trained on CIFAR-10.

As expected, the difference in accuracy here is less stark than the case of ResNet. This is because of two reasons - 1. The dimension of the activation layer in ResNet before the last ResNet block is 16,384 whereas the activations before the third last FC layer is only 512. 2. Figure 5 shows that the difference in the variance ratio between the LR and the N-LR network is much smaller as compared to Figure 1 for ResNets.

### C.3 CLASS WISE VARIANCE

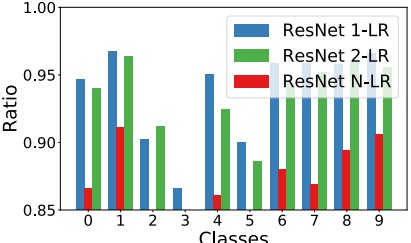

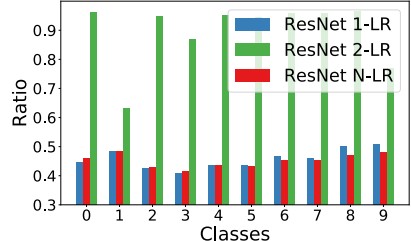

(a) 512 dimensional activations from after last ResNet block.

(b) $16k$ dimensional activations from before last ResNet block.

Figure 6: Class wise variance ratio of one singular values for the activations before the last ResNet block.

In this experiment, we plot $\text{VR}_1(\cdot)$ for embeddings of examples restricted to individual classes. Figure 6(a) shows the variance ratio captured by the largest singular value for the activations before the last FC layer while Figure 6(b) shows the variance ratio captured by the largest singular value for the activations before the last ResNet block. These experiments give us some idea about the extent to which the set of basis vectors assigned to individual classes are intersecting.

### C.4 CLUSTERS OF LOW DIMENSIONAL EMBEDDINGS

Figure 7 shows the two dimensional projections of the 2048 dimensional embeddings obtained from ResNet-50-LR and ResNet-50-N-LR. The coloring is done according to the coarse labels of the input. We can see that the clusters are more separable in the case of the model with LR-layer than the model without, which gives some insight into why a max-margin classifier performs better for the LR model than the N-LR model.

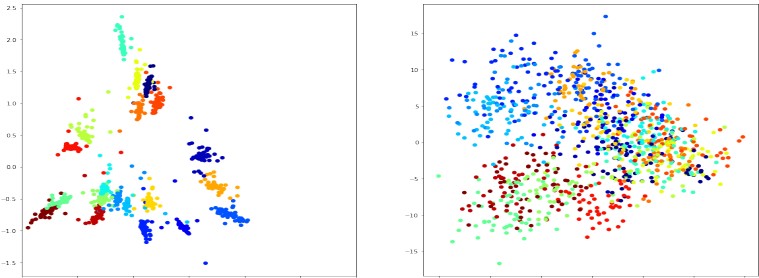

Figure 7: PCA plot for super-class labels in CIFAR 100. Plot on the left shows embeddings from the LR trained model on ResNet-50 while the plot on the right represents a normal ResNet-50 model. Both of them are trained in a similar way. Each color represents a different class.

# D ADVERSARIAL ATTACKS

## D.1 TYPES OF ATTACKS

- Iter-FSGM- The Fast Sign Gradient Method (FSGM) Goodfellow et al. (2014) was proposed as the existing methods (Szegedy et al. (2013)) of the time were slow. FSGM tries to maximize the loss function by perturbing the input slightly. Iterative Fast Sign Gradient method (Iter-FSGM) is a simple extension of FSGM that follows the following simple iterative step.

$$\mathbf{x}_a^0 = \mathbf{x}_d, \tag{4}$$
$$\mathbf{x}_a^{n+1} = \text{clip}_{\mathbf{x},\epsilon}(\mathbf{x}_a^n + \alpha \cdot \text{sign}(\nabla_{\mathbf{x}_a^n}\mathcal{L}(\mathbf{x}_a^n, \mathbf{y}_t)))$$

- Iter-LL-FSGM- Iter-FSGM is an untargeted attack. Iterative less likely fast sign gradient method (Iter-LL-FSGM) (Kurakin et al., 2017) is a way to choose the target label wisely. Consider $\mathbb{P}_M(\mathbf{y}|\mathbf{x})$ to be the probability assigned to the label $\mathbf{y}$, for the example $\mathbf{x}$, by the model $M$. In this attack, the target is set as $\mathbf{y}_t^n = \text{argmin}_{\mathbf{y} \in \mathcal{Y}} \mathbb{P}_M(\mathbf{y}|\mathbf{x}^n)$ and the following iterative update steps are performed.

$$\mathbf{x}_a^0 = \mathbf{x}_d, \tag{5}$$
$$\mathbf{x}_a^{n+1} = \text{clip}_{\mathbf{x},\epsilon}(\mathbf{x}_a^n - \alpha \cdot \text{sign}(\nabla_{\mathbf{x}_a^n}\mathcal{L}(\mathbf{x}_a^n, \mathbf{y}_t^n)))$$

Intuitively, this method picks the least likely class in each iteration and then tries to increase the probability of predicting that class. In both of these methods, $\alpha$ was set to 1 as was done in Kurakin et al. (2017).

- DeepFool- Moosavi-Dezfooli et al. (2016) describes the DeepFool procedure to find the optimal (smallest) perturbation for the input $\mathbf{x}$ that can fool the classifier. In the case of affine classifiers, DeepFool finds the closest hyper-plane of the boundary of the region where the classifier returns the same label as $\mathbf{x}$ and then adds a small perturbation to cross the hyper-plane in that direction.

  As deep net classifiers are not affine, the partitions of the input space where the classifier outputs the same label are not not polyhedrons. Hence, the algorithm takes an iterative approach. Specifically, the algorithm assumes a linerization of the classifier around $\mathbf{x}$ to approximate the polyhedron and then it takes a step towards the closest boundary. For a more detailed explanation please look at Moosavi-Dezfooli et al. (2016).

## D.2 MINIMUM PERTURBATION FOR A SUCCESSFUL ATTACK

Table 3 lists the minimum perturbation required to fool the classifier under the particular attack scheme. For Iter-FSGM and Iter-LL-FSGM, there are essentially three hyper-parameters$(t, \alpha, \epsilon)$ in the experiments as can be seen below.

Iter-FSGM

**Repeat $t$ times** $\tag{6}$

$$\mathbf{x}_a^0 = \mathbf{x}_d,$$
$$\mathbf{x}_a^{n+1} = \text{clip}_{\mathbf{x},\epsilon}(\mathbf{x}_a^n + \alpha \cdot \text{sign}(\nabla_{\mathbf{x}_a^n}\mathcal{L}(\mathbf{x}_a^n, \mathbf{y}_t)))$$

Iter-LL-FSGM

**Repeat $t$ times** $\tag{7}$

$$\mathbf{x}_a^0 = \mathbf{x}_d,$$
$$\mathbf{x}_a^{n+1} = \text{clip}_{\mathbf{x},\epsilon}(\mathbf{x}_a^n - \alpha \cdot \text{sign}(\nabla_{\mathbf{x}_a^n}\mathcal{L}(\mathbf{x}_a^n, \mathbf{y}_t^n)))$$

Following the convention of Kurakin et al. (2017), we set $\alpha = 1$. We tuned the hyper-parameter $\epsilon$ function to obtain the smallest $\epsilon$ that resulted in over 99% misclassification accuracy for some $t$ and then repeated the experiments until such a $t$ was achieved. Finally $\rho$ was calculated.

Algorithm 2 in Moosavi-Dezfooli et al. (2016) gives details about the DeepFool algorithm for multi-class classifiers. The algorithm returns the minimum perturbation $r(\mathbf{x})$ required to make the classifier misclassify the instance $\mathbf{x}$. The $L_2$ dissimilarity is obtained by calculating $\rho = \frac{r(\mathbf{x})}{\|\mathbf{x}\|_2}$

For the benefit of reproducibility of experiments, we list the values of $\epsilon$ for Iter-LL-FSGM and Iter-FSGM in Table 6 corresponding to the values in Table 3 . For DeepFool, we used the publicly available code [6].

|  | Model | $\epsilon$[Iter-LL-FSGM ] | $\epsilon$ [Iter-FSGM ] |
|---|---|---|---|
|  | ResNet **2-LR** | 0.04 | 0.02 |
| White Box | ResNet **1-LR** | 0.06 | 0.01 |
|  | ResNet **N-LR** | 0.01 | 0.01 |
| Black Box | ResNet **1-LR** | 0.08 | 0.01 |
|  | ResNet **2-LR** | 0.1 | 0.01 |

Table 6: Value for $\epsilon$ required for Adversarial Misclassification corresponding to Table 3.

### D.3 UNSTABILITY OF ADVERSARIAL ATTACKS

The essential difference between the attacks in Figure 2 and Table 3 is in the number of iterations for which the updates (Step 6 and Step 7) are executed. In Figure 2, the step is executed $t$ times whereas in Table 3, the updates are executed until the classifier makes a mistake.

It would be natural to expect that once a classifier has misclassified an example, adding more adversarial perturbation will surely not make the classifier classify it correctly. However, Figure 8 suggests that a misclassified example can be classified correctly upon further addition of noise.

Let $y_a(\mathbf{x}; k)$ be the label given to $\mathbf{x}$ after adding adversarial perturbation to $\mathbf{x}$ for $k$ steps. We define *instantaneous accuracy* $(a_{\mathcal{I}}(k))$ and *cumulative accuracy* $(a_{\mathcal{C}}(k))$ as

$$a_{\mathcal{I}}(k) = 1 - \frac{1}{m} \sum_{i=1}^{m} \mathcal{I}_{0,1} \left\{ y_a(\mathbf{x}; k) \neq y_a(\mathbf{x}; 0) \right\}$$

$$a_{\mathcal{C}}(k) = 1 - \frac{1}{m} \sum_{i=1}^{m} \max_{1 \leq j \leq k} \left\{ \mathcal{I}_{0,1} \left\{ y_a(\mathbf{x}; j) \neq y_a(\mathbf{x}; 0) \right\} \right\}$$

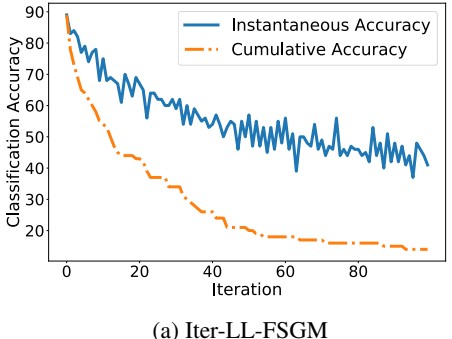

(a) Iter-LL-FSGM

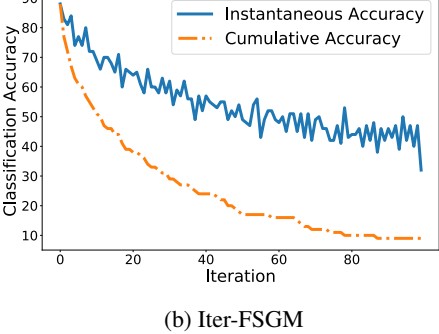

(b) Iter-FSGM

Figure 8: This shows that an adversarial example that has successfully fooled the classifier in a previous step can be classified correctly upon adding more perturbation. Figure **??** and **??** refers to the two attack schemes - Iter-LL-FSGM and Iter-FSGM respectively.

---

[6]https://github.com/LTS4/DeepFool/blob/master/Python/deepfool.py

In Figure 8, we see the *instantaneous accuracy* and the *cumulative accuracy* for ResNet 1-LR where $\alpha = 0.01, \epsilon = 0.1$ and $t$ is plotted in the x-axis. The cumulative accuracy is by definition a non-increasing sequence. However, surprisingly the instantaneous accuracy is not monotonic and has a lower rate of decrease than the cumulative accuracy. It also appears to stabilize at a value much higher than the cumulative accuracy.

### D.4 ADVERSARIAL ATTACK ON MAXIMUM MARGIN MODEL

Here, we train max-margin classifiers on representations of images obtained from different ResNet models and see whether the representations of adversarial images, that had successfully fooled the ResNet model, could fool the max-margin classifier as well. We train a variety of X-MAXG models - ResNet18-1-LR-MAXG, ResNet18-2-LR-MAXG, ResNet18-N-LR-MAXG and black box versions of the same.

Once, these classifiers are trained, adversarial examples were generated for the three attacks (both black box and white box) on ResNet18-1-LR, ResNet18 2-LR and ResNet18-N-LR by the techniques described in Section D.2. The accuracy of the MAXG models are listed in Table 4.

To perform a fair comparison with ResNet18-N-LR-MAXG, it is essential to add a similar amount of noise to generate the examples for ResNet18-N-LR-MAXG as is added in ResNet18-1-LR-MAXG. The adversarial examples are hence generated by obtaining the gradient using ResNet18-N-LR but stopping the iteration only when the adversarial example could fool ResNet18-1-LR. This is, in-fact, the black box attack on ResNet18-1-LR. As Table 4 suggests, our models create representations that are more robust to adversarial perturbations than a normal model.

