# OpenReview forum: "Intriguing Properties of Learned Representations"
_ICLR.cc/2019/Conference_

### Official Review · AnonReviewer1 · 2018-11-01
**A relatively novel idea but poorly written paper**

**Rating:** 5
**Confidence:** 2

**Review:**

This paper introduces a new way to have more compressed (lower rank) representation of the data in a supervised fashion. The authors motivates their work by saying that such representation are more useful for transfer learning and are more robust to adversarial examples! In order to achieve this, the authors introduce a virtual LR layer and utilize Nystrom technique to make the process more efficient.  The idea introduced in this paper is interesting but the paper is poorly written and organized which makes its through evaluation difficult. Below, I provide more detailed comments.

I don't understand at what frequency the low-rank optimization as the  subproblem to Equation (OPT) is being done. Given that the DNN training  is being done using batched of examples, where do you put the low-rank  optimization, in the end of each epoch? It seems it is in the end of each epoch but it should be clearly stated.

I don't understand motivation for L_N(A). The authors justify it by  saying that a trivial solution for the optimization problem would be setting A+b=0 and they introduce this term to avoid it. However, this is not correct. Note that there are n examples in A and we can only make one of them  zero at a time. (Note that talking about A+b is not also accurate  because they are not of the same size).

In order to use Nystrom method for low-rank approximation, W needs to be  a symmetric postive semi-definite matrix. I am not sure if the heuristic  procedure introduced in Page 4 is well-justifed. and whether we are  still optimizing the objective function introduced in Page 3. Do we  still have a stable training?

What is ResNet N-LR in experiment? The authors introduced ResNet 1-LR and ResNet 2-LR but not ResNet N-LR! I found the description N-LR later but the naming is rather confusing. I would use LR instead of N-LR because it seems it has N LR layer. I would also explain this next to other methods.

Not sure if I understand Bottle-LR. The description in the text is not clear and I don’t understand the motivation for this baseline! Again, this should be described next to other methods.

The authors do not mention what is their setting for r in the experiments (Table 1).

Page 5, Paragraph after Table 1: First CIFAR-100 should be CIFAR-10.

One way the authors defend their framework is to have a representation that can be used in transfer learning. Nonetheless, the results in Table 1.b shows that their framework is not doing good for transfer learning.

Not sure if I understand Figure 1. How do you change the number of singular values? I understand this is a hyper parameter for your framework but I am confused how it is being set for N-LR method. Similarly, I don’t understand Table 2 and how you change the embedding dimension.


Unlike the claim made by the authors, it seems that VGG19 N-LR does better compared to VGG19 2-LR in Figure 2.

---

> ### Author Response · Authors · 2018-11-22
> **Reply to Reviewer 1**
>
> [Norm Constrained Loss]
> -- A + b does not have to be exactly zero but only by being very small, L_c (reconstruction loss) can be orders of magnitude smaller than L (classification loss), while retaining full rank activations. Also, the magnitude of A+b can be decreased to an arbitrarily small amount  without changing the output of the network. Here are two ways to make that happen:
>
> Let the structure be X --(W_1)--> W_1X (Say A) --(W_2)--> W_2X . You can make W_1 of the order 10^-4 to reduce the magnitude of A and then increase the magnitude by the inverse of the amount ~~ 10^4 and then get a similar magnitude of W_2X.
> In case of most state of the networks (eg. one with ReLU), the scale of the weights is immaterial as if all the parameters can be multiplied by a positive constant c, due to positive homogeneity,  the output before the last layer will be multiplied by a constant c^k for some k. As the final layer is just a linear classifier, the classification output will not change.
>
> [A+b]
> -- This is clarified in page 13 footnote 3.
>
> [Frequency of rank optimization]
> --  No, it is not done at the end of the epoch. It is done every 10 iterations. We will mention it in the revised paper in Appendix C with other experimental details. As the SVD is very cheap in this case, it can be done frequently. The algorithm does not minimize the rank of activations of each batch (as done in Leczema et. al 2017) but rather learns a low rank subspace where the activations finally lie it is learnt continuously.
>
> [Heuristic for Nystrom]
> --  Though the problem is being changed (and problems are often changed by a small amount to make it easier to solve), a solution of this changed problem is also a solution of the original problem as the activations have to be low rank to solve this changed problem.
>
> [N-LR]
> --The last line of paragraph 3 in Page 5 describes N-LR as “we use the term N-LR to represent models without LR-layers”.
>
> [Bottle-LR]
> -- We have added more description in Section 3 and hope that it explains why it is comparable as a benchmark.
>
> [Setting of target rank r]
> --In page 5 paragraph 3, we refer to Appendix C for further details on the experimental procedure and in Appendix C we mention “The target rank for the LR-layers, which were placed before the FC layers was set to 50 while the target rank for the LR-layer before the last ResNet block was set to 1000”
>
> [Problem with Cifar100 in Page 5]
> --We do not see anything amiss there.
>
> [Transfer Learning]
> -- We do not claim that our model is a better way of doing transfer learning. We say that ”the LR model suffers a small loss of 4% in accuracy as compared to the N-LR model” in the task of transfer learning. The point of this exercise is to show that by reducing the rank of the representations we are not removing useful information from the representations, which isn’t directly useful for the current task (predicting on coarse labels). It shows that information is still preserved to a significant extent even after decreasing the rank and this information shows is used in Figure 1b for the transfer learning exercise.
>
> [Understanding Figure 1]
> --- The number of singular values in Figure 1 is not a hyper-parameter. It is a quantity we measure by the equation of Variance Ratio described in the first sentence of Section 3.
>
>
> [VGG19 in Figure 2]
> --We are extremely sorry for the mistake in the labelling. We have now uploaded the correct legends and we hope it will solve your confusion about this.

---

### Official Review · AnonReviewer3 · 2018-11-06
**Interesting augmentation of training procedure to induce low-rank activations at intermediate layers, but unable to evaluate the significance**

**Rating:** 6
**Confidence:** 2

**Review:**

Synopsis:
Overall, this paper was fairly well written and seems to have an original approach towards inducing low-rank structure on the space of activations in some intermediate layer in a computationally efficient way without changing the underlying model. This training modification does not seem to affect test performance and the low-rank embeddings that are learned seem useful at discriminative tasks. Adversarial robustness also appears improved.

Pros:
--While I am not familiar enough with the background literature on model compression in neural networks, I thought the augmented optimization problem used to induce low-rank structure on the space of activations was interesting and worthy of investigation. The authors appear to get great results in Table 1 & Table 2.

Cons:
--I cannot really gauge the significance of the result against other existing approaches towards low-dimensional representations because of my limited familiarity with the relevant literature. However, I didn’t feel quite convinced by the discussion in the paper that low-rank activations were superior to other kinds of low-rank approximations, for instance to the network weights (c.f. discussion in Appendix A). I think the discussion on this topic could be a bit improved.
--With respect to the writing, I’m a bit uncertain as to the primary message of the paper. While it seems to introduce a new augmented training approach for generating compressed representations which potentially has practical utility, based on the paper title and scattered discussion it seems to suggest that the representations themselves are interesting, e.g. the idea of having low-rank activations while largely maintaining test performance. I didn’t fully understand the extent to which the results are intriguing or helpful in understanding neural networks. Could this be developed a bit more?

Miscellaneous comments:
--In Figure 2, the accuracy with respect to adversarial perturbations seems to drop more for VGG19 2-LR (pink curve) than the model VGG19 N-LR (brown), which seems counter to your point on robustness?
--In Figure 4, why is the behavior of ResNet 2-LR (blue curve) similar to ResNet N-LR (red curve)? I would’ve expected any number of LR layers to increase the sensitivity in intermediate layers to adversarial input perturbations.

---

### Official Review · AnonReviewer2 · 2018-11-06
**Too few experiments for many analyses**

**Rating:** 3
**Confidence:** 4

**Review:**

This paper presents a way to induce low-rank representations in a deep neural network and study its effect on adversarial attacks.

Quality

The analyses are conducted on several types of problems, first classification tasks for confirmation of the low-rank structure, and then on adversarial attacks. Unfortunately, there is only a very few number of experiments per analysis, making it virtually impossible to infer reliably any trends in the data. Table 1, 2, 3 and 4 contains at best enough information for a proof of concept, but it is not possible to make any conclusion out of them. Also, VGG results only appear in figure 2 where they appear to contradict the conclusions held by the authors. Is it difficult to understand why results from VGG do not appear in any table.

Clarity

The paper is difficult to follow. Introduction gives too much details and even contains methodological information, all of which obfuscates the main message which does get more clear in the latter sections. The sections 2 and 3 are confusing because they do not follow the logic presented in the abstract. The latter states that observations on the low-rank structure of the representations will be done prior to experimentally impose low-rank. However, section 2 presents the low-rank structure imposed on models while section 3 presents the observations of low-rank-representations jointly with the results of imposed low-rank structure.

There is no clear definitions of what the "intriguing properties" are beside the fact that forced low-rank representations yield similar results on classification and are more robust to adversarial attacks on a very limited number of experiments.

Originality

Using low-rank representation is not something new and has already been explored in [1] for instance.

Significance

There would be an important contribution to make if the author would analyze the effect of low-rank by varying the constraint. However, the current analyses are not pushed far enough to get any useful insight using only a fixed rank and making a minor modification by adding one or two LR-layers. Experiments in table 2 is a good step in this direction nonetheless.

[1] Luo, Ping. "Learning deep architectures via generalized whitened neural networks." In International Conference on Machine Learning, pp. 2238-2246. 2017.

---

> ### Author Response · Authors · 2018-11-22
> **Reply to Reviewer 2**
>
>  [Title]
> -- By intriguing properties, we refer to the property that representations tend to lie in a moderately low rank affine space, with our method we show that further enforcing this allows for compression with little loss of accuracy and adversarial robustness. However will change the title of the paper to reflect the outcome of the discussion of the rebuttal.
>
> [Originality (Comparison with [3])]
>
>
> X is the N x d data matrix (eg. original images).
> Z is the N x m matrix of learned representations (i.e. after the activations).
>
> We focus on ensuring that Z is low rank whereas most prior work focuses on making the weight matrices low rank[1,2,3]. Note that low rank weight matrices do not imply low rank for Z. (cf. Appendix A).
>
> [3] is significantly different from our work as it  does not ensure low rank representation, which is our core contribution. It designs low rank whitening matrix P from data correlation \Sigma to speed up the training, using SGD, of ‘Whitening NNs’ [5].
>
> As opposed to learning low rank representations, the paper multiplies the representations  with a low rank whitening matrix  to get a whitened representation. As the whitening matrix is low rank, it can be constructed from \Sigma using a fast SVD and this reduction in complexity is the main point of the paper. (They also claim improvement in generalization).
> Regarding the algorithm, the whitening matrix is generated from the data and not “learnt” during the training and more crucially, there is no evidence in the paper that the learned networks produces any low rank representations. On the other hand we designed an algorithm that ensures that the representations of our trained network will lie in a low rank affine space.
> Also, the rank constraints are much more aggressive in our case e.g. in ResNet 2-LR it is of the order of 1/100 as opposed to ½ in [3]. Our papers are fundamentally different and having a low rank whitening matrix during training and having a network which learns to produce low rank representations after training are very different.
>
> The only similar work that we have found is [3] and has been referenced in the paper while highlighting the differences in algorithm and analysis between the two papers.
>
> [Significance]
>
> -- In Page 5, paragraph 3 addresses this point. Specifically the sentence “As a side note, we observed that the target rank is not an essential hyper-parameter as the training enforces a much lower rank than what is set.” shows that varying the constraint does not have any significant effect on the training. [4] is a contemporary work in CVPR’17(referred to in the paper) and it also motivates a similar idea of low rank activations. Experimentally, it shows superior classification performance on different datasets and is an example of another useful applications of low rank activations.
>
> [VGG Experiments]
>
> -- We are very sorry for the mixup in the legends. The revised paper has the correct legends. We will include the results for VGG for some more experiments.
>
>
> [1] Jaderberg, Max, Andrea Vedaldi, and Andrew Zisserman. "Speeding up convolutional neural networks with low rank expansions." arXiv preprint arXiv:1405.3866 (2014).
> [2] Denton, Emily L., et al. "Exploiting linear structure within convolutional networks for efficient evaluation." Advances in neural information processing systems. 2014.
> [3] Luo, Ping. "Learning deep architectures via generalized whitened neural networks." In International Conference on Machine Learning, pp. 2238-2246. 2017.
> [4] Lezama, José, et al. "OLE: Orthogonal low-rank embedding, a plug and play geometric loss for deep learning." The IEEE Conference on Computer Vision and Pattern Recognition (CVPR). 2018.
> [5] Desjardins, Guillaume, Karen Simonyan, and Razvan Pascanu. "Natural neural networks." Advances in Neural Information Processing Systems. 2015.

---

> > ### Comment · AnonReviewer2 · 2018-11-26
> > **Reply**
> >
> > Comparison with [2]
> >
> > The whitening matrix of [2] is a reparametrization of the representation. It is not forcing to learn a low-rank representation, it is forcing any learnable representation to be low-rank through the reparamatrization. In that sense, it is very comparable to this work.
> >
> > The counter example of appendix A is based on the assumption that the low-rank is forced on the weights, inducing a low-rank projection before the activation. Then, the counter example show that the non-linear function of the activation can affect the rank of the representation, sometimes increasing it. This counter example is not valid for [2], as the low-rank is forced on the representation, prior to the affine transformation. What I argue is that this low-rank transformation in [2] should be viewed as the representation itself and not as a transformation on the representation. The relation between whitened NN and KFAC explained in [1] suggests that the representation of a whitened NN should be viewed as PZ, not Z.
> >
> > Low-rank Analyses
> >
> > If the low-rank target has barely any effect because training already enforces a much lower rank than the target, then why would the proposed constraint make any difference to training?
> >
> > [1] Desjardins, Guillaume, Karen Simonyan, and Razvan Pascanu. "Natural neural networks." Advances in Neural Information Processing Systems. 2015.
> > [2] Luo, Ping. "Learning deep architectures via generalized whitened neural networks." In International Conference on Machine Learning, pp. 2238-2246. 2017.

---

> > > ### Author Response · Authors · 2018-11-30
> > > **Reply**
> > >
> > > Comparison with [2]]
> > >
> > > We thank the reviewer for pointing us to the suggestion in  [1] that the representation of a whitened NN should be viewed as the PZ  i.e. the whitened representations and not Z i.e. the original representation. Due to the low rank whitening matrix in [2], the whitened representations in [2] are indeed low rank though the original representations might not be.  However, [2] does not study it from the point of view of obtaining low rank representations and does not observe any of the analyses we do.
> > >
> > > Simply doing a PCA on the representations and using the low rank projection matrix(PCs) to obtain a low rank representation does not even given a good enough test classification score (Table 2) and hence, does not make sense to be used for adversarial testing.  Our benchmarking algorithm Bottle-LR also learns a low rank projection matrix and our experiments show the difference in its behaviour with our model in Figure 2 and Table 1b.
> > >
> > >
> > > [Low Rank Analyses]
> > >
> > > Figure [1a] shows the difference between the approximate low rank induced by normal training and the stronger low rank imposed by our algorithm. Specially Figure 1b shows the difference our algorithm can create.
> > >
> > > Regarding "why would the proposed constraint make any difference to training", along with the difference in the low rank structure in Figure 1, Table 2(a), 2(b), 3, 4 and Figure 2, 3 and 4 also show the other differences introduced by our constraint.

---

> > > > ### Comment · AnonReviewer2 · 2018-11-30
> > > > **Reply to authors**
> > > >
> > > > Comparison with [2]
> > > >
> > > > The method proposed in this paper is indeed different than [2], the proposed one being in the form of a penalty while the latter is a reparametrization. There is nonetheless a strong similarity and a comparison with [2] should be done, even if they do not study it from the same perspective. This would arguably be a better benchmark than the bottleneck layers in Bottle-LR.
> > > >
> > > > [Low Rank Analyses]
> > > >
> > > > Results in the paper does suggest a difference in behavior when using the low rank penalty. It is not clear however why such difference appears. If the reason is the lower rank of the representation, then why wouldn't the different target values affect the magnitude of change compared to the baseline? The current experiments test whether the low-rank penalty does induce a low-rank structure, but not if the low-rank structure is the reason why it induces better embeddings for classification or make networks more robust to adversarial attacks. If the sole reason is the low-rank structure, then any method inducing equivalent low-rank structure would have a similar effect, hence the importance of comparing with [2].
> > > >
> > > > Conclusion
> > > >
> > > > In short, because there is not enough experiments per analysis to identify trends reliably and because the experiments only test whether it works rather than why it works, we both don't know if it generalizes to other problems and don't understand well why it works in the current ones. Nevertheless, I believe the subject seems promising and I would encourage the authors to continue on this line of work regardless of the outcome of this submission.

---

### Meta-Review · Area_Chair1 · 2018-12-13
**Interesting topic but the analysis is lacking**

**Confidence:** 4
**Recommendation:** Reject

**Metareview:**

Dear authors,

The reviewers all appreciated the interest of studying properties of the latent representations rather than of the weights. The impact of the rank on the robustness to adversarial attacks is also of interest.

There were, however, two main issues raised. Due to the lack of confidence of some reviewers, I reviewed the paper myself and found the same issues:
- Clarity could be improved. Some models are mentioned before being described (N-LR) and some important details are missing. In particular, we sometimes lose track of the goal of the experiments. For instance, there are quite a few experiments on the further reduction of the rank of the representation but it is not clear what to extract from them.
- More importantly, there are several important gaps in the analysis. In particular: a/ As many reviewers have pointed out, low-rank constraints on the weight matrices induce low-rank representations if the activation function is linear. As it is not, this might not be true but deserves a discussion. b/ You state that the rank constraint has little effect given that the actual rank is much less than the constraint. However, one would expect the resulting rank to be a smooth function of the rank of the constraint. Since there is a discrepancy between ResNet N-LR and ResNet 1-LR, this should be investigated. c/ For the robustness to black-box adversarial attacks, these attacks are constructed using the N-LR models. Is is thus not too surprising that those models do not perform as well.

Thus, despite the lack of confidence of one reviewer (the question about the N-LR models might stem from the fact that it is used before being introduced), I strongly encourage you to take their comments into account for a future submission.